# In-Line Aerosol Therapy via Nasal Cannula during Adult and Paediatric Normal, Obstructive, and Restrictive Breathing

**DOI:** 10.3390/pharmaceutics15122679

**Published:** 2023-11-27

**Authors:** Marc Mac Giolla Eain, Ronan MacLoughlin

**Affiliations:** 1Research and Development, Science and Emerging Technologies, Aerogen Ltd., Galway Business Park, H91 HE94 Galway, Ireland; 2School of Pharmacy and Biomolecular Science, Royal College of Surgeons in Ireland, D02 YN77 Dublin, Ireland; 3School of Pharmacy and Pharmaceutical Sciences, Trinity College, D02 PN40 Dublin, Ireland

**Keywords:** drug delivery, aerosol therapy, vibrating mesh nebuliser, adult, paediatric, high-flow nasal therapy, obstructive and restrictive lung disease, breath pattern

## Abstract

High-flow nasal oxygen therapy is being increasingly adopted in intensive and home care settings. The concurrent delivery of aerosolised therapeutics allows for the targeted treatment of respiratory illnesses. This study examined in-line aerosol therapy via a nasal cannula to simulated adult and paediatric models with healthy, obstructive and restrictive lung types. The Aerogen Solo vibrating mesh nebuliser was used in combination with the Inspired^TM^ O2FLO high-flow therapy system. Representative adult and paediatric head models were connected to a breathing simulator, which replicated several different states of lung health. The aerosol delivery was quantified at the tracheal level using UV-spectrophotometry. Testing was performed at a range of supplemental gas flow rates applicable to both models. Positive end-expiratory pressure was measured pre-, during and post-nebulisation. The increases in supplemental gas flow rates resulted in a decrease in aerosol delivery, irrespective of lung health. Large tidal volumes and extended inspiratory phases were associated with the greatest aerosol delivery. Gas flow to inspiratory flow ratios of 0.29–0.5 were found to be optimum for aerosol delivery. To enhance aerosol delivery to patients receiving high-flow nasal oxygen therapy, respiratory therapists should keep supplemental gas-flow rates below the inspiratory flow of the patient.

## 1. Introduction

Oxygen therapy has become the first-line treatment option for patients suffering respiratory distress in both critical and home care settings. It can be used with infants, children and adults to treat numerous respiratory conditions, such as asthma [1], sleep apnoea [2], cystic fibrosis (CF) [3], chronic obstructive pulmonary disease (COPD) [4,5], COVID-19 [6] and numerous others. High-flow nasal oxygen therapy (HFNO) delivers heated and humidified oxygen at high flows, which can exceed the patient’s inspiratory flow demand. It allows for the accurate control of the inspired oxygen fraction, reduced work of breathing, induces a positive end-expiratory pressure (PEEP) that enables alveolar recruitment and has been reported to aid in the clearance of carbon dioxide from the nasopharyngeal dead space [7,8,9,10].

Aerosol therapy, concurrent to HFNO, has become an increasingly popular treatment option. Surveys of clinical practice show that 64% of adult [11] and up to 70% of paediatric [12] respiratory therapists deliver aerosol therapy concurrent to HFNO. Patients who are removed from HFNO for conventional aerosol therapy may have prolonged periods without the prescribed oxygen supplementation and pressure support. For those who use the nebuliser with a facemask or mouthpiece over the nasal cannula, reduced amounts of aerosol are delivered [13,14]. The outcomes arising from sub-optimal aerosol therapy have been suggested to include an increased length of stay and time to a reduction of oxygen supports [15]. For these reasons, as well as ease of use, tolerance of patients and control of gas flow rate, the use of an in-line nebuliser—delivering aerosol through the same circuit as the carrier gas—is the most logical and attractive approach to aerosol therapy concurrent to HFNO.

Previous in vitro studies that have combined aerosol delivery with HFNO systems have examined factors such as nebuliser type [16], placement in the circuit [17], humidification effects [18], nasal cannula size and design [19], supplemental gas flow rate [20] and delivery gas type [21]. Vibrating mesh nebulisers (VMN) are favoured amongst clinicians [11,12]. In the single head-to-head radio scintigraphy study reported in humans, Dugernier et al. [16] reported superior delivery with an in-line VMN over an in-line JN, 3.6% versus 1%. Furthermore, JN themselves are contraindicated against by the largest HFNO therapy system manufacturers for reasons including interference with both oxygen FiO2 and gas flow rates, and require the circuit to broken for JN refilling, which may lead to lung derecruitment, a reduction in the applied PEEP and release of fugitive medical and bioaerosols [22,23,24].

The HFNO gas flow rate is one of the primary determinants of aerosol delivery, with lower flow rates and larger bore cannula facilitating greater aerosol delivery [14,25,26,27]. Ari [28] proposed that the gas flow rate should be matched with the patients’ inspiratory flow to maximise aerosol deposition, while Li et al. [29] noted a higher and more consistent aerosol dose at gas flow rates less than half of a patient’s inspiratory flow. Interestingly, the combination of both the nebuliser position and gas flow rate has recently been shown to influence the relative ratio of aerosol deposition within the upper airways and lungs. While in a scintigraphy study using an in-line VMN, Alcoforado et al. [14] recorded lung deposition levels of 17.23% and upper air-way deposition of 34.5% at 10 LPM, decreasing to 3.5% and 46.1%, respectively, at 50 LPM. The HFNO system itself could have an impact on aerosol delivery [17]. Often the main determinant directly influencing aerosol transit through the system is the humidifier pot and circuit type. The placement of the nebuliser on either the dry or wet side of the humidifier and the use of heated humidified circuits have been described with varying amounts of aerosol reported to be lost in the humidifier water pot and through impaction on the circuit tubing walls [18,30].

Naturally, the condition of the patient will affect the quantity of aerosol available for inhalation during HFNO. A number of in vitro studies [20,21,30] have examined the effects of the breathing pattern on aerosol delivery during HFNO therapy. The primary focus of these studies has been distressed breathing patterns, which are associated with larger tidal volumes, ~750 mL, and rapid breath rates, ~30 BPM. Unsurprisingly, the respirable mass of aerosol was significantly higher during simulated distressed breathing compared to quiet breathing. However, to date, few studies have examined the influence of lung health, healthy versus diseased lung states (restrictive and obstructive lung diseases), where not only the tidal volume, breath rate and inspiratory-to-expiratory (I:E) ratio vary but also airway resistance and compliance, on aerosol delivery during HFNO. This work addresses these gaps in the literature.

The purpose of this study was to examine the effects of lung health, normal versus diseased, in simulated adult and paediatric models receiving in-line aerosol therapy via nasal cannula concurrent to high-flow nasal oxygen therapy. Additionally, for the first time, we report on the use of a low-flow mode (Inspired^TM^ O2FLO, Vincent Medical, Kowloon, Hong Kong), which provides an easy-to-use interface for the selection of a lower gas flow rate for an extended time period, which may provide ease of use benefit for concurrent aerosol therapy.

## 2. Materials and Methods

### 2.1. Nebuliser

All testing was completed with a VMN (Aerogen Solo, Aerogen Ltd., Galway, Ireland) and its associated controller (Aerogen Pro-X, Aerogen, Ltd., Galway, Ireland). The performance characteristics of the VMN were measured by laser diffraction (Malvern Instruments, Malvern, UK) as previously reported. Ref. [31] and was defined in terms of the volume mean diameter (Dv50) or average aerosol droplet size, 4.20 ± 0.06 µm and aerosol output flow rate, 0.49 ± 0.02 mL/min, respectively.

### 2.2. High-Flow Nasal Therapy System

The Inspired^TM^ O2FlO (Vincent Medical, HK) system, Figure 1, was used. The VMN was incorporated into the system using the standard Aerogen 22 mm adult T-piece (Aerogen Ltd., Galway, Ireland), which was placed on the wet side of the humidifier pot. An adult integrated heated breathing circuit (PN: 510-049D) was used with an adult nasal cannula (Large, PN: 51006179). A paediatric integrated heated breathing circuit (PN: 510-050D) with a paediatric nasal cannula (size medium, PN: 51005243). Testing was conducted across a range of gas flow rates for both models, 10, 30 and 50 LPM adult and 3 and 7 LPM paediatric. The Inspired^TM^ O2FLO system is equipped with a “Low Flow Mode”, which provides a pre-set from 5 to 30-min interval therapy function, at from 6 to 30 LPM gas flow, for a physician’s convenience. The low flow mode of operation was set to 6 LPM for 20 min to facilitate easy selection of the lower, more aerosol-compatible gas flow rate.

### 2.3. Characterisation of Aerosol Dose

Figure 1 is a schematic illustration of the experimental apparatus used to characterise the dose of aerosol available at the level of the trachea. Testing was completed on both simulated adult and paediatric models. The models are based on scans of the nose–throat region (nasal cavity, pharynx and larynx) of a 53-year-old adult male and 5-year-old female [20,32,33]. The relevant nasal cannula was positioned in the nose of the applicable head model as per the manufacturer’s guidelines. The head models were connected to a breathing simulator (ASL 5000, IngMar Medical Inc., Pittsburgh, PA, USA) via a collection filter (Respirgard 303, Vyaire, Dublin, Ireland) positioned at the level of the trachea. The drug collected on this capture filter was washed using 10 mL of de-ionised water and quantified using UV Spectrophotometry. Drug recovery using this specific method was 100 +/− 5%. Three different states of lung health for both model types were simulated—healthy, obstructive and restrictive, see Table 1. The profiles used in this study should not be considered as definitive and can vary significantly in patients depending on severity of the condition. The simulated breathing profiles were generated from the patient profile library provided by the breathing simulator manufacturer and were derived from the published literature, see [34]. The flow dynamics of the generated breathing profiles, including the peak inspiratory flow, were measured with a flow analyser (CITREX H5, IMT Analytics, Buchs, Switzerland) placed between the capture filter and the breathing simulator.

A 2.5 mL dose of 5 mg/2.5 mL Albuterol Sulphate (TEVA Pharmaceuticals, Waterford, Ireland) was placed in the nebuliser’s medication cup and aerosolised. The tracheal dose was determined by measuring the quantity of aerosol captured on the collection filter via UV-spectrophotometry (WPA, Lightwave II, Biochrom Ltd., Cambridge, UK) at 276 nm and interpolated on a standard curve of Albuterol Sulphate concentrations, 3.125–200 µg/mL. The results are presented as the mean ± standard deviation of the nominal dose placed in the nebuliser’s medication cup. All testing was completed in independent quintuplicate (n = 5). There were no recorded instances of HFNO system or nebuliser malfunction or alarms throughout the entire experimental series. Time to delivery of the dose was noted to be 5 min +/− 15 s for all runs.

The positive end-expiratory pressure (PEEP) (cm H2O) was measured pre-, during and post-nebulisation using the flow analyser (CITREX H5, IMT Analytics, Buchs, Switzerland). This was placed between the capture filter and the breathing simulator. Measurements were continuously recorded at 0.1 s intervals.

## 3. Results

### 3.1. Tracheal Dose

Figure 2 is a plot comparing the effects of supplemental gas flow rate and lung health on the tracheal dose in simulated spontaneous breathing (a) adult and (b) paediatric models. Irrespective of lung health, as the supplemental gas flow rate increases the tracheal dose (%) decreases in both models. The simulated healthy lung received the largest tracheal dose at the lower flow rates, 16.58 ± 0.36% healthy, 7.70 ± 0.51% obstructive and 15.12 ± 0.54% restrictive at 6 LPM in the adult. However, as the supplemental gas flow rates above 10 LPM, the restrictive and obstructive conditions received larger doses of aerosol, 0.73 ± 0.13% healthy, 0.90 ± 0.25% obstructive and 1.20 ± 0.23% restrictive at 30 LPM, 0.38 ± 0.09% healthy, 0.50 ± 0.15% obstructive and 0.23 ± 0.06% restrictive at 50 LPM. Across the spectrum of flow rates examined in the paediatric model, the model with the obstructive lung disease received the largest dose of aerosol, followed by healthy and then restrictive, see Figure 2b.

Table 2 compares the unique low-flow delivery mode of the O2FLO with a standard low supplemental gas flow rate of 10 LPM. Student’s *t*-tests were performed to determine the significant difference in a tracheal dose between the two supplemental flow rates for each state of lung health. There was no significant difference in tracheal deposition, with *p* >> 0.05 for each case.

### 3.2. Delivery at Peak Inspiratory Flow

Table 3 presents the peak inspiratory flow for the different breathing profiles considered for the simulated adult models. Included in the table is the tracheal dose (%) recovered at the peak inspiratory flow.

Figure 3 shows the relationship between the tracheal dose (%) (mean ± SD) and the gas flow to inspiratory flow (GF:IF) ratio across the range of flow rates and lung profiles included in the adult part of this study. The tracheal dose (%) for lung profile peaks between 0.29 and 0.5. GF:IF > 0.5 results in a significant decrease in the tracheal dose (%) (mean ± SD).

### 3.3. Positive End-Expiratory Pressure

Table 4 and Table 5 compare the positive end-expiratory pressure (PEEP) (cm H_2_O) pre-, during and post-nebulisation at different supplemental gas flow rates (LPM) for different lung types in an adult, Table 4, and paediatric, Table 5, simulated spontaneous breather. A one-way ANOVA with post-hoc Tukey tests was completed to determine significance, which was considered at *p* < 0.05. In the adult models, Table 4, there was a single instance where there was a significant change in PEEP measured at 6 LPM obstructive *p* = 0.005. Across all other flow rates and simulated lung conditions, there was no significant change in the PEEP, *p* >> 0.05. While in the paediatric models, Table 5, there were also no statistically significant changes in PEEP pre-, during or post-nebulisation at all flow rates and lung conditions considered, *p* >> 0.05.

## 4. Discussion

This study examined the use of in-line aerosol therapy administered during HFNO in spontaneously breathing adult and paediatric models. Both simulated cohorts had varying lung conditions—obstructive and restrictive—and the quantity of aerosol that could potentially reach the lungs was benchmarked with healthy lungs. The effects of in-line aerosol therapy on PEEP were also measured across the spectrum of conditions examined in this study.

It has long been accepted that as the supplemental gas flow rate increases, the quantity of aerosol available at the lungs decreases [13,19,21,29]. The differences in reported values between this study, from 0.38 ± 0.09% to 15.99 ± 0.71% adult and from 0.52 ± 0.26% to 4.07 ± 0.45% paediatric, and others can be attributed to differences in the breathing pattern [25,30], head model [35], HFNO system [16] and cannula sizes [36]. However, the findings presented in this work, for both models irrespective of lung health, follow the same trends as these and numerous other published works.

Unsurprisingly, variations in lung condition—healthy, restrictive, obstructive—were found to have a statistically significant effect on the aerosol available at the tracheal level, *p* << 0.05, see Appendix A. The different states of lung health are associated with variations in tidal volume, breath rate, I:E ratio, compliance and resistance. Réminiac et al. [30] and Dailey et al. [21] compared the effects of breathing patterns, quiet versus distressed, on the quantity of aerosol available in the lungs. Distressed breathing, with a larger tidal volume and higher breath rate, was found to yield a greater delivered dose, 10.3% versus 6.7%, 6.7% versus 3.5% and 5.1% versus 3% at flow rates of 30, 40 and 50 LPM [30]. Bennett et al. [25] examined the effects of tidal volume, breath rate and I:E ratio variations on the tracheal dose (%) in simulated adult and paediatric models receiving in-line aerosol therapy with HFNO. Increases in tidal volume, breath rate and extended inspiratory phases were associated with greater tracheal doses in both groups. Bennett et al. [25] found that the optimal I:E ratio in adults was 1:1.15 and 1:1 in paediatrics. In a similar study, Bauer et al. [37] concluded that the longer inspiratory time facilitated an increase in aerosol delivery, while losses were minimised with a rapid expiratory phase. Our data are consistent with these findings, Figure 2a,b and Appendix A. In the simulated adult model, the healthy and restrictive breathing patterns generated almost double the tidal volume of the obstructive condition and had I:E ratios of 1:1. However, unlike in the works of [21,25,30], this trend was not observed above 10 LPM. We hypothesise that this is due to the differences in the various head models, HFNO systems and cannulas and simulated profiles used across those studies.

Unlike in the adult models, the paediatric model with the largest tidal volume, quickest breath rate and longest inspiratory phase was the model with the obstructive lung disease, followed by healthy and then restrictive, see Table 1. Our findings, presented in Figure 2b) and Appendix A, are consistent with this—the simulated obstructive model generated the largest dose of aerosol at the tracheal level (%) followed by healthy and then restrictive.

Ari [28] proposed that aerosol delivery concurrent to HFNO would be maximised by matching the supplemental gas flow (GF) rate with the patient’s inspiratory flow (IF), or GF:IF = 1. Our data, Figure 3, shows that the optimum aerosol delivery occurs at GF:IF < 1. Peak tracheal dose (%) was measured across the range 0.29–0.5 and decreased as GF:IF increased. At low supplemental gas flow rates, there is less turbulent flow created within the HFNO system and in the model airways; thus, less aerosol is lost due to impaction within the unit, aerosol particle coalescence and deposition in the upper airways. This results in more aerosol available at the tracheal level. This is consistent with works by Li et al. [29] and Réminiac et al. [30], although they found that the peak tracheal dose (%) extended down as low as 0.1. This might be explained by differences in the head model, HFNO system and breath settings. In a survey conducted by Li et al. [11] to establish the worldwide clinical practice of HFNO with concurrent aerosol therapy in adults, only ~30% of respondents reported reducing the supplemental gas flow rate during aerosol therapy. Our findings indicate that supplemental gas flow should be reduced to levels below a patient’s peak inspiratory flow rate to ensure optimum aerosol delivery.

Maintaining open patent airways is key to maximising the delivery of aerosol within the lungs themselves, as well as for normal lung function. As such, it is important to maintain an optimal PEEP during aerosol delivery [38,39]. To the best of the authors’ knowledge, this is the first study that has measured PEEP pre-, during and post-nebulisation concurrent to HFNO therapy, demonstrating that the vibrating mesh nebuliser operation does not affect the applied PEEP to a patient during aerosol therapy. Statistical analysis of the data comparing PEEP pre-, during and post-nebulisation showed only one instance in the adult study where there was a significant difference in PEEP. This occurred in the obstructive model at 6 LPM. This is most likely due to the breathing type—extended expiratory phase and high compliance, and the consistency in the data—very small standard deviations.

There are several limitations to this study. The use of capture filters at the tracheal level results in an overestimation of the delivered dose (%) as the filters do not allow exhalation of the aerosols that are not deposited. The results should not be considered absolute; rather, the focus should be on the trends in the data. The breathing patterns used in this study should not be considered definitive and can vary significantly depending on the severity of the disease. Further work is required to expand on these breathing profiles and lung models. Only a single nasal cannula size and design were included for each model. There are several sizes available for both adults and paediatric patients. Future studies examining the effects of cannula size and design are required to expand on the findings in this study. The level of leak between cannula and nares, as well as the newer designs of asymmetrical nasal cannula may provide some further insight into seeking out gains in aerosol delivery, balanced with patient comfort and safety. The study was completed with atmospheric air, 21% oxygen, as the carrier gas. In a clinical setting, depending on the condition of the patient, it might be necessary to use an oxygen mixture to generate a higher FiO_2_. Future work is required to determine the effects, if any, of higher oxygen concentrations on aerosol delivery concurrent to HFNO therapy.

## 5. Conclusions

This in vitro study examined the use of in-line aerosol therapy concurrent with HFNO therapy in simulated adult and paediatric models with differing states of lung health. Increases in supplemental gas flow rate resulted in a decrease in the quantities of aerosol available for inhalation at the lungs in both groups, irrespective of lung health. Large tidal volumes and extended inspiratory phases were associated with greater tracheal doses (%) in both model groups. GF:IF ratios between 0.29 and 0.5 were found to be the optimum ratio for in-line aerosol delivery during HFNO therapy. The measurements of PEEP were taken pre-, during and post-nebulisation during HFNO therapy in both model groups. No significant change in PEEP was measured in the paediatric models. One variation was found in the adult model at the lowest flow rate with a simulated obstructive lung disease, which was most likely due to the extended exhalatory phase. This article will be of considerable benefit to those seeking to implement in-line aerosol therapy concurrent with HFNO.

## Figures and Tables

**Figure 1 pharmaceutics-15-02679-f001:**
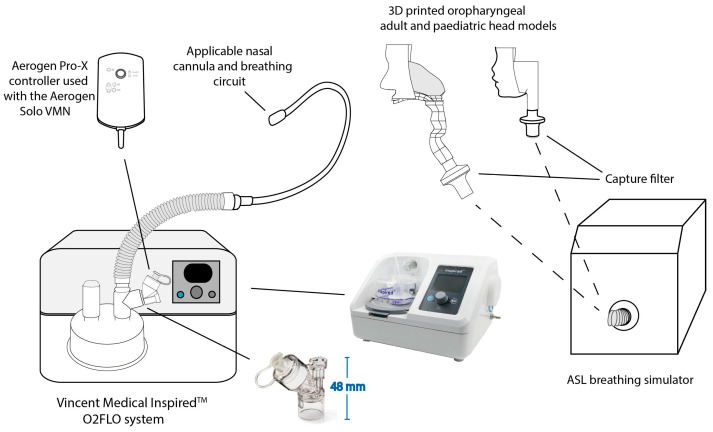
Schematic illustration of the experimental test setup.

**Figure 2 pharmaceutics-15-02679-f002:**
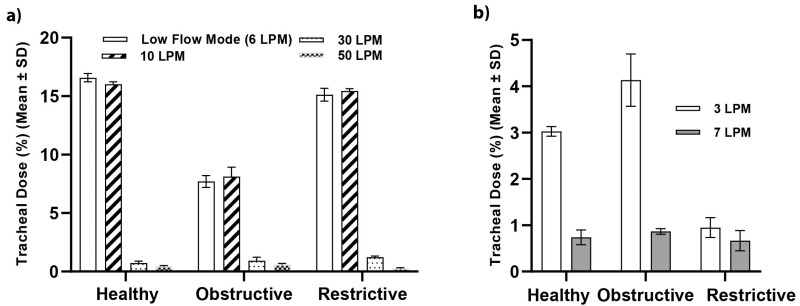
Comparison of the effects of supplemental gas flow rate and lung health on the tracheal dose (%) (mean ± SD) in a simulated (**a**) adult and (**b**) paediatric spontaneously breathing model.

**Figure 3 pharmaceutics-15-02679-f003:**
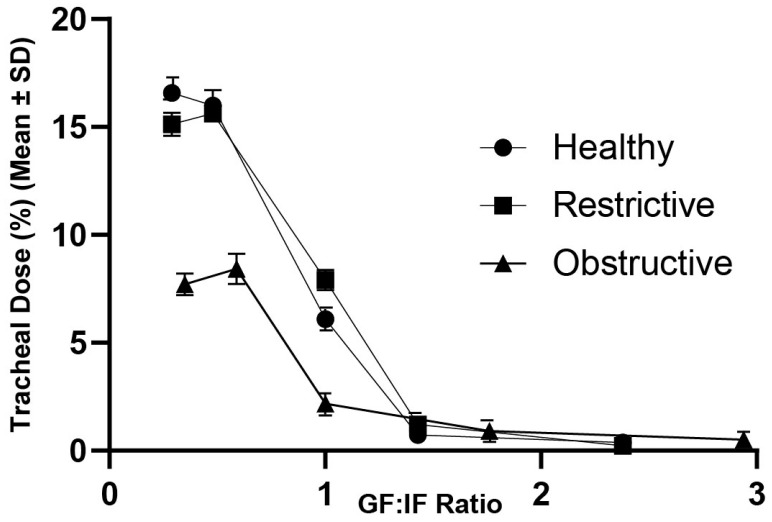
Plot of tracheal dose (%) (mean ± SD) variations with changes in GF:IF ratio and lung health for a spontaneously breathing model adult.

**Table 1 pharmaceutics-15-02679-t001:** Breathing patterns and lung states of the representative adult and paediatric models simulated. GF = gas flow rate. IF = inspiratory flow rate.

	Adult			Paediatric		
	*Healthy*	*Obstructive*	*Restrictive*	*Healthy*	*Obstructive*	*Restrictive*
**Breath Rate (BPM)**	15	22	15	20	20	30
**Tidal Volume (mL)**	500	270	500	300	350	70
**I:E Ratio**	1.0:1.0	1.0:2.5	1.0:1.0	1.0:2.0	1.0:1.8	1.0:2.85
**Compliance (L/cmH_2_O)**	0.05	0.080	0.04	0.01	0.08	0.05
**Resistance (cmH_2_0/L/S)**	5	20	20	30	5	25
**GF:IF ratio**	0.29–2.38	0.35–2.94	0.29–2.38	-	-	-

**Table 2 pharmaceutics-15-02679-t002:** Comparison of the low-flow mode with a standard low supplemental gas flow rate for different states of lung health. *p* < 0.05 denotes a statistically significant difference.

	Low Flow Mode (6 LPM) (Mean ± SD)	10 LPM (Mean ± SD)	*p*-Value
**Healthy**	16.58 ± 0.36	15.99 ± 0.71	0.344
**Obstructive**	7.70 ± 0.51	8.42 ± 0.70	0.253
**Restrictive**	15.12 ± 0.54	15.63 ± 0.30	0.573

**Table 3 pharmaceutics-15-02679-t003:** Tracheal dose (%) (mean ± SD) at peak inspiratory flow in the simulated adult with different states of lung health.

	Adult	
	Peak Flow (LPM)	Tracheal Dose (%) (Mean ± SD)
**Healthy**	21	6.09 ± 0.18
**Restrictive**	21	7.90 ± 0.48
**Obstructive**	17	2.17 ± 0.21

**Table 4 pharmaceutics-15-02679-t004:** Comparison of positive end-expiratory pressure (PEEP) (cm H_2_O) pre-, during and post-nebulisation in a simulated spontaneously breathing adult for changes in supplemental gas flow rate (LPM) and lung health. *p* < 0.05 denotes a significant difference.

Low Flow (6 LPM)				
	PEEP (cm H_2_O)			
**Lung Health**	*Pre-nebulisation*	*During*	*Post nebulisation*	*P-value*
**Healthy**	0.113 ± 0.003	0.122 ± 0.009	0.115 ± 0.012	0.472
**Restrictive**	0.065 ± 0.002	0.072 ± 0.002	0.065 ± 0.005	0.077
**Obstructive**	0.130 ± 0.006	0.163 ± 0.011	0.158 ± 0.008	0.033
**10 LPM**				
**Healthy**	0.139 ± 0.004	0.141 ± 0.007	0.147 ± 0.006	0.084
**Restrictive**	0.069 ± 0.009	0.061 ± 0.007	0.060 ± 0.002	0.086
**Obstructive**	0.246 ± 0.008	0.264 ± 0.009	0.267 ± 0.009	0.005
**30 LPM**				
**Healthy**	0.740 ± 0.033	0.767 ± 0.029	0.773 ± 0.028	0.227
**Restrictive**	0.512 ± 0.016	0.525 ± 0.014	0.527 ± 0.011	0.240
**Obstructive**	0.790 ± 0.041	0.810 ± 0.032	0.814 ± 0.026	0.510
**50 LPM**				
**Healthy**	1.785 ± 0.098	1.825 ± 0.046	1.853 ± 0.041	0.308
**Restrictive**	1.279 ± 0.117	1.306 ± 0.078	1.322 ± 0.063	0.749
**Obstructive**	1.525 ± 0.091	1.552 ± 0.070	1.574 ± 0.058	0.593

**Table 5 pharmaceutics-15-02679-t005:** Comparison of positive end-expiratory pressure (PEEP) (cm H_2_O) pre-, during and post-nebulisation in a simulated spontaneously breathing paediatric for changes in supplemental gas flow rate (LPM) and lung health. *p* < 0.05 denotes a significant difference.

3 LPM				
	PEEP (cm H_2_O)			
**Lung Health**	*Pre-nebulisation*	*During*	*Post nebulisation*	*P-value*
**Healthy**	5.010 ± 0.067	5.032 ± 0.375	5.334 ± 0.085	0.075
**Restrictive**	0.899 ± 0.045	0.923 ± 0.061	0.935 ± 0.052	0.565
**Obstructive**	5.345 ± 0.290	5.620 ± 0.294	5.579 ± 0.289	0.308
**7 LPM**				
**Healthy**	8.710 ± 1.177	9.137 ± 1.590	9.073 ± 1.649	0.888
**Restrictive**	3.841 ± 0.431	4.056 ± 0.296	4.089 ± 0.244	0.464
**Obstructive**	10.254 ± 0.371	10.528 ± 0.108	10.661 ± 0.259	0.089

## Data Availability

The data presented in this study are available in this article (and Appendix A).

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
