# Peer review of "In-Line Aerosol Therapy via Nasal Cannula during Adult and Paediatric Normal, Obstructive, and Restrictive Breathing"

_pharmaceutics, 2023, doi:10.3390/pharmaceutics15122679_

Round 1

Reviewer 1 Report

Comments and Suggestions for Authors

Please, find comments in the attached file. 

Reviewer 2 Report

Comments and Suggestions for Authors

Review report of the manuscript pharmaceutics-2713399

by M. M. Giolla Eain and R. MacLoughlin

The work is interesting and makes a contribution to the field of nasal aerosol delivery. The authors present an in vitro study of a 3D printed nasal cavity and trachea submitted to nasal drug therapy. Aerosol deposition was obtained and evaluated in the phantom. Drug deposition within the airways was discussed with the existen literature data.

In general the procedures presented in the study are ok. However, it is necessary to clarify and extend the section Materials and methods.

1. The profiles mentioned in the lines 137-139 shoul be provided for sake of clarity. the authors refers to the literature. However, it is necessary to explain more in details this aspect.

2. In view of the previous ponit, it is recommended to accurately describe the meaning of 'obstructed' and 'restricted' airways. Is it only about different flow rates? no geometrical modiciations?

3. Please describe clearly the boundary conditions of the experimental set-up, using a Figure if necessary.

4. Have you consider the resistance of the non printed airways? Your model is truncated, hence, which condition is given at the tracheal section? This condition could be crucial for the respiratory patterns and, as a consequence, for the aerosol deposition...

5. Please state clearly which is the purpose of the study at the end of the section 'Introduction'.

Round 2

Reviewer 2 Report

Comments and Suggestions for Authors

The authors have responded to all the concerns highlighted in the review report so that I am pleased to recommend the manuscript for publication.